# Impact of Aerobic Capacity on Mechanical Variables in Track Sprinters and Middle-Distance Runners: A Comparative Study

**DOI:** 10.3390/jfmk10030342

**Published:** 2025-09-09

**Authors:** Nikolaos P. Belechris, Gregory C. Bogdanis, Elias Zacharogiannis, Athanasios Tsoukos, Giorgos Paradisis

**Affiliations:** School of Physical Education and Sports Science, National and Kapodistrian University of Athens, 17237 Athens, Greece; nikosbele@phed.uoa.gr (N.P.B.); gbogdanis@phed.uoa.gr (G.C.B.); elzach@phed.uoa.gr (E.Z.); atsoukos@phed.uoa.gr (A.T.)

**Keywords:** anaerobic speed reserve, force–velocity, repeated sprints

## Abstract

**Background:** This study examined the impact of aerobic capacity on force–velocity (F–v) variables and repeated-sprint (RS) performance in male national-level sprinters (SPRs, *n* = 8; 177.0 ± 4.3 cm; 74.0 ± 5.0 kg; maximal oxygen uptake [VO_2_max]: 55.4 ± 3.0 mL/kg/min) and middle-distance runners (MDRs; *n* = 8; 179.0 ± 5.1 cm; 67.2 ± 5.0 kg; VO_2_max: 64.3 ± 3.3 mL/kg/min). **Method:** Participants underwent assessments of aerobic capacity, mechanical F-v profiling in sprinting 2 × 60 m with full recovery, and a 10 × 60 m repeated-sprint test with 30 s recovery. **Results:** MDRs exhibited significantly higher VO_2_max (*p* < 0.001) and speed at VO_2_max (vVO_2_max, *p* < 0.001), while SPRs demonstrated greater anaerobic speed reserve (ASR, *p* < 0.001), maximal theoretical horizontal force (F_0_, *p* = 0.012), and power output (P_max_, *p* < 0.01). During the RS test, SPRs displayed a 16.6% performance decrement (*p* = 0.002) and failed to complete all sprints with voluntary withdrawal after 5–8 sprints due to exhaustion, whereas MDRs maintained consistent performance. SPRs exhibited a larger decrease in v_0_ compared to MDRs (*p* < 0.01), whereas no differences were observed on F_0_ (*p* = 0.519) and P_max_ (*p* = 0.758). Blood lactate accumulation was higher in SPRs (*p* < 0.001). Multiple linear regression analysis on the pooled sample identified vVO_2_max (*p* = 0.003) and not ASR (*p* = 0.482) as a key predictor of fatigue resistance. **Conclusions:** These findings underscore the critical role of aerobic capacity in sustaining RS performance. Aerobic capacity, specifically vVO_2_max, emerged as the primary determinant of fatigue resistance during repeated sprints, underscoring its critical role in sustaining RS performance over mechanical variables such as v_0_ but not F_0_ and P_max_.

## 1. Introduction

Repeated-sprint (RS) ability, i.e., the ability to perform multiple maximal sprints with minimal performance decline, is essential in intermittent sports such as soccer, rugby, and track events, reflecting anaerobic power and rapid recovery, making it a key focus of training to enhance athletic performance [1,2]. Key mechanical variables—force (F), velocity (v), and power (P)—underpin sprint performance and correlate strongly with maximum running speed (v_max_) [3,4]. In the horizontal F–v profile, F_0_ is the theoretical maximal horizontal force (at zero velocity) and v_0_ the theoretical maximal running velocity (at zero force), and P_max_ is the maximal horizontal power derived from these values. RF_max_ is the maximal ratio of horizontal to total force, and DRF is the rate at which this ratio declines as running speed increases. Changes in these parameters during RS efforts reveal the neuromuscular system’s fatigue resistance [5,6].

Fatigue-induced declines in RS are driven by metabolic and neuromuscular factors—including phosphocreatine resynthesis, metabolite clearance, and muscle fiber composition—which directly affect mechanical variables and sprint consistency [7,8]. Additionally, they arise from peripheral mechanisms, impaired excitation–contraction coupling, and reduced sarcolemmal excitability, while central factors like diminished voluntary activation may also contribute to fatigue-induced declines in RS [9]. Monitoring mechanical variables, such as v_0_ and P_max_ shifts across sprints, thus offers insights into both anaerobic capacity and recovery kinetics, since they correlate highly with sprint performance [10]. However, the mechanical variables mentioned have not been explored during RS along with aerobic tests. Aerobic capacity, typically evaluated by VO_2_max, is a critical determinant of RS performance, since faster PCr resynthesis and metabolite clearance reduce fatigue and sustain sprint consistency [11,12,13,14]. Yet VO_2_max alone does not fully explain endurance or RS outcomes, while vVO_2_max is often a better predictor of endurance performance [15,16]. In other words, VO_2_max reflects the maximal volume oxygen an athlete can consume per unit of time (min), whereas vVO_2_max is the running speed at which this maximal oxygen uptake is reached, providing a practical link between aerobic capacity and performance.

The anaerobic speed reserve (ASR), defined as the difference between v_max_ and vVO_2_max, reflects the balance of anaerobic and aerobic capacities and has been shown to be vital for middle-distance performance [17]. However, to our knowledge, only limited research has examined the relationship between ASR and RS. Moreover, blood lactate measurements provide a sensitive and reliable indicator that the athletes exercised at maximal intensity and estimate the corresponding metabolic demand.

Sprinters (SPRs) and middle-distance runners (MDRs) use distinctly different energy system profiles during training with SPRs typically prioritizing anaerobic power, whereas MDRs emphasize aerobic endurance, which likely alter their RS-related mechanical variables in v_0_, F_0,_ and P_max_. However, the extent to which these training adaptations and differences in ASR affect F-v parameters during fatigue in RS tests remains unclear. Comparing SPRs and MDRs on RS mechanics will clarify the role of aerobic capacity, sprint speed, and ASR, thus informing targeted training interventions. Therefore, the purpose of this study was to compare the impact of aerobic capacity and sprint speed on the F-v parameters during an RS test performed by track SPRs and MDRs. We hypothesized that fatigue during RS, expressed as a reduction in sprint performance and F-v parameters, would be lower in MDRs, compared to SPRs, especially P_max_ and v_0_ [18], due to their higher aerobic capacity.

## 2. Materials and Methods

### 2.1. Participants

An a priori power analysis using G*Power 3.1 for a repeated measures ANOVA (within–between interaction) with an effect size f = 0.3, α = 0.05, power (1−β) = 0.95, 2 groups, 10 measurements, and an assumed correlation among repeated measures of 0.5 indicated that a total sample size of 16 participants is required. Sixteen male athletes, eight 400 m SPRs (age: 20.9 ± 2.13 years, 400 m race time: 50.7 ± 1 s) and eight 800 m MDRs (age: 20.6 ± 2.38 years, 800 m time: 115.12 ± 3.84 s, 400 m training time: 52.2± 1.47 s), participated in this study. All participants were runners with a minimum of five years of training experience at a competitive level, including participation in national championships and were free from lower extremity injury for at least 3 months prior to testing. Additionally, they were trained runners who regularly performed all the tests used in this study as part of their annual training routine. The experimental procedures were approved by the Ethics Committee of the School of P.E. and Sport Science, National and Kapodistrian University of Athens (approval protocol number: 1204/1507-2020), in accordance with the Declaration of Helsinki. All participants gave their written informed consent to participate and agreed to avoid any strenuous type of training during the study period.

### 2.2. Experimental Design

Participants performed three tests (VO_2_max, F-v profile, and RS ability) at least 48 h apart. Participants were instructed to refrain from intense physical activity, alcohol, and caffeine consumption for at least 24 h before each testing session. A standardized warm-up was performed before each test, and all runners were fully familiarized with the testing procedures, as these were regularly incorporated into their training and assessment routines.

### 2.3. VO_2_max Test

Participants performed an incremental test for the determination of VO_2_max, vVO_2_max, and the ventilatory threshold (VT). Following a 5 min warm-up at 10 km/h, treadmill (Technogym Runrace 1200, Italy), velocity was increased by 1 km/h every 2 minutes until volitional fatigue. This protocol had been validated from other studies for the determination of VO_2_max, VT, and vVO_2_max simultaneously [19]. Gas collection was made during the last 30 s period of each 2 min stage to allow the athlete to attain steady-state oxygen consumption (VO_2_). VO_2_ was measured by the open-circuit Douglas bag method. The athlete breathed through a low-resistance 2-way Hans-Rudolph 2700 B valve. The expired gases passed through a 90 cm length and 340 mm diameter flexible tubing into 150 L capacity Douglas bags. The concentrations of carbon dioxide (CO_2_) and O_2_ in the expired air were measured using an Electrolab, FerMac 368, UK Carbon Dioxide and Oxygen Analyzer. The gas analyzer was calibrated against standardized gases (15.88% O_2_, 3.95% CO_2,_ and 100% nitrogen). Expired air volume was measured by means of a dry gas meter (Harvard) previously calibrated against standard airflow with a 3 L syringe. Barometric pressure and gas temperature were recorded, and respiratory gas exchange data [i.e., VO_2_, volume of carbon dioxide (VCO_2_), minute ventilation (VE)] were determined for each workload on a locally developed computer program based on the computations described by McArdle, Katch, and Katch when VE is at ambient temperature and pressure, saturated with water vapor, and both the fractional concentration of carbon dioxide in expired air (FECO_2_) and fractional concentration of oxygen in expired air (FEO_2_) are known [20]. The highest VO_2_ value obtained during the incremental exercise test was recorded as the subject’s VO_2_max, which also elicited a heart rate within ± 10 bpm of age-predicted maximum heart rate (HRmax), respiratory exchange ratio (RER) greater than 1.05, and a rating of perceived exertion greater than 19 in the 20-grade Borg scale. Also, vVO_2_max was defined as the treadmill speed corresponding to the highest VO_2_ value measured via the Douglas bag method during the incremental protocol. All raw gas exchange data were used without filtering, as measurements were recorded over stable 30 s intervals at the end of each 2 min stage by the same technician, to minimize inter-rate variability.

### 2.4. Horizontal F–v Profiling

Horizontal F-v profile was assessed by performing two maximal 60 m sprints, starting from a two-point start. One high-speed camera (Casio EX-F1, Tokyo, Japan; sampling frequency 300 fps) was continuously recording the entire sprint distance. The camera was mounted on stable tripods and was placed in the middle of the sprint distance (i.e., at 30 m). Thirteen marking poles were placed at adjusted positions along the 60 m distance to determine 5 m split distances while avoiding parallax error [5]. The sprint acceleration mechanical variables were obtained using the method of Samozino [21] based on an inverse dynamic approach applied to the body’s center of mass. In this method, horizontal external force, velocity, and power were obtained from time data measured during the acceleration phase of each sprint, every 5 m for the total 60 m distance. From the horizontal force and sprint velocity values, individual force–velocity relationships were determined using least-squares linear regression. F_0_ and v_0_ were then identified as the x- and y-intercepts of the force–velocity relationships, respectively, and P_max_ was calculated as F_0_ × v_0_/4. Their fastest 60 m sprint was used to derive their individual F-v parameters. Anaerobic speed reserve (ASR) was calculated as the difference between an athlete’s v_max_ and their vVO_2_max [22]. The formula used for ASR calculation is as follows:$$ASR=vmax-{\mathrm{vVO}}_{2}\max$$

Repeated-Sprint Test: A 10 × 60 m RS test with a 30 s rest interval between each sprint was performed to measure changes in sprint performance and alterations in the mechanical F-v variables for each sprint [21]. Performance decrement was calculated using the following formula [23]: $$\text{Slope Decrement}=[(\frac{\text{slope over completed sprints}}{\text{fastest time}})\times 100]\times (\text{Completed sprints}-1)$$

The slope of the linear regression between sprint performance time and sprint number was multiplied by 4–9 (i.e., number of sprints—1) to express the percent decline in performance from the first to the last sprint. All participants were instructed to sprint all efforts maximally and verbal encouragement was provided to minimize pacing strategies.

Blood lactate concentration was measured every 2 sprints with a Lactate Plus Meter (L+, Nova Biomedical, Waltham, MA, USA), which uses an electrochemical lactate oxidase biosensor for the measurement of lactate in whole blood. A blood sample of 0.7 μL was required; sample analysis time was 13 s. Test strips used with the L^+^ do not require calibration codes or specific calibration strips. The L^+^ was supplied with two levels of a quality control solution (level 1: 1.0–1.6 mM; level 2: 4.0–5.4 mM) that was used prior to testing to ensure correct operation of the analyzer. Due to the short recovery time, measurements were taken 10 s after the end of the sprint. 

### 2.5. Statistical Analysis

Data are presented as mean ± standard deviation (SD). Student’s t-test for independent samples was used to compare anthropometric and performance characteristics between the two groups. MANOVA was used to examine aerobic variables and sprint variables to account for their intercorrelations and to control the overall Type I error rate, followed by univariate ANOVAs to identify which specific variables differed between groups. Normality of each dependent variable was confirmed via Shapiro–Wilk tests (all *p* > 0.05). Assumptions of sphericity for the repeated measures ANOVAs were tested using Mauchly’s test, and when violations were detected, the Greenhouse–Geisser correction was applied to adjust the degrees of freedom. Two-way repeated measures ANOVAs (SPRs vs. MDRs × 10 sprints) were used to compare F–v and sprint performance variables. For participants who did not complete all 10 sprints, only the available sprint data were included in the analysis (missing sprint values were omitted rather than imputed). This approach allowed for the examination of the main effects of groups and number of sprints, as well as potential interactions between groups and the number of sprints. Post hoc analyses were conducted using Bonferroni-adjusted pairwise comparisons test to identify specific differences between groups and across number of sprints when significant main or interaction effects were detected. Multiple linear regression models were employed to examine the relationships between fatigue, vVO_2_max, and ASR. Multicollinearity, normality, and homoscedasticity among predictors were assessed using correlation coefficients, variance inflation factors, Shapiro–Wilk, Breusch–Pagan, and ncvTest, indicating no issues. Statistical significance was set at *p* < 0.05 for all analyses except MANOVAs, for which—based on the number of dependent variables—alpha was set at 0.05 divided by the number of dependent variables. Effect sizes were calculated using Cohen’s d and partial η^2^ to quantify the magnitude of observed differences. Cohen’s d provides a standardized measure of effect size for pairwise comparisons, with values interpreted as small (d = 0.2), medium (d = 0.5), or large (d = 0.8) effects. The partial eta squared (partial η^2^), used in the context of ANOVA, is interpreted as small (0.01), medium (0.06), or large (0.14) effects, providing an indication of the strength of the association between factors and the outcome.

## 3. Results

### 3.1. Anthropometric Characteristics and Aerobic Capacity

Table 1 summarizes the anthropometric and performance characteristics of the two groups, for which no significant differences were observed in height or body fat. The t-test did not reveal a statistically significant difference between the two groups for v_max_, although the result had a trend towards significance and the effect size was large (*p* = 0.052, d = –1.03). This may suggest that the observed change may be relevant for performance despite it not being statistically significant. The 400 m race times of the SPRs were significantly different from the 400 m training times of the MDRs (t (14) = –2.42, *p* = 0.03, d = 1.21). The inclusion of 400 m training times for MDRs was used to ensure comparable sprinting experience between groups, providing a relevant benchmark for performance comparisons.

A one-way MANOVA was conducted on the three aerobic variables (VO_2_max, vVO_2_max, VT) with athlete group as the independent factor, using a Bonferroni-adjusted alpha of 0.05/3 = 0.0167. The omnibus test was significant (Pillai’s Trace = 0.904, F(3, 8) = 25.09, *p* = 0.0002), indicating a large multivariate effect of group. Follow-up univariate ANOVAs revealed significant group differences for VO_2_max (F(1, 10) = 25.19, *p* = 0.0005; d = 2.86), vVO_2_max (F(1, 10) = 84.40, *p* < 0.00001; d = 4.04), and VT (F(1, 10) = 54.50, *p* = 0.00002; d = 3.18), all reflecting large effect sizes.

### 3.2. F–v Profiles

A MANOVA was conducted for sprint variables (F_0_, v_0_, P_max_, RF_max_, DRF) with group being the independent variable. This analysis did not reveal a significant multivariate effect, suggesting no overall differences in the sprint profiles between groups (Table 2). It should be noted that the statistical significance threshold for the MANOVA was adjusted to *p* < 0.01 (i.e., 0.05/5). The univariate ANOVAs did not show significant group differences for any of the variables.

### 3.3. Repeated-Sprint Performance and Fatigue 

Figure 1 shows sprint performance during the RS test for the MDRs and SPRs. As shown in Figure 1, SPRs were able to complete only 5–8 sprints due to voluntary exhaustion. The two-way ANOVA revealed significant main effects of group (F (1, 117) = 12.54, *p* < 0.001, partial η^2^ = 0.05), number of sprints (F (9, 117) = 5.54, *p* < 0.001, partial η^2^ = 0.30), and interactions (F (7, 117) = 4.27, *p* < 0.001, partial η^2^ = 0.20). The results of the post hoc analysis revealed significant differences in sprint performance time across several sprints for the SPR. Specifically, comparisons between the 1st sprint vs. 2nd sprint showed significant differences (Δ = 0.72 s, *p* < 0.05), indicating a decrease in sprint performance time, thus further supporting the notion of decreased sprint performance time during repeated sprints (Figure 1). For the MDRs, there were no statistically significant differences between the sprints, indicating consistent sprint performance time throughout the RS.

To compare the effect of fatigue on RS performance and the difference between the two groups, the fatigue slope decrement was calculated as the percentage decrement in performance across all completed sprints. SPRs showed a mean slope decrement of 16.66% ± 7.02%, whereas MDRs showed a mean decrement of 2.55% ± 6.14%. The difference was statistically significant with a large effect size (*p* < 0.001, d = −2.14).

### 3.4. Changes in Mechanical Variables

Repeated ANOVAs of the F–v variables were conducted to analyze how each variable was affected by fatigue during the repeated sprints. For v_0_, the results revealed significant main effects for the groups (F (1, 117) = 21.15, *p* < 0.001, partial η^2^ = 0.11) and the number of sprints (F (9, 117) = 4.20, *p* < 0.001, partial η^2^ = 0.24), as well as a significant interaction (F (7, 117) = 2.56, *p* = 0.017, partial η^2^ = 0.13). The estimated marginal means for the MDRs indicate small fluctuations (7.82 m/s to 8.26 m/s). In contrast, for the SPR group, they indicate a broader range of fluctuations across the sprints (9.31 m/s to 7.88 m/s). Significant differences were observed between the 1st sprints for the MDR and SPR group, with a mean speed difference of 1.05 m/s (*p* < 0.001). The within-participant coefficient of variation (CV) of v_0_ was significantly higher in SPRs than in MDRs (mean CV: 8.87% vs. 3.70%, t (8.30) = −4.79, *p*< 0.001, d = 2.39), indicating substantially higher intra-individual variability in v_0_ across repeated sprints in SPRs.

For F_0_, the ANOVA did not reveal significant main effects of group or number of sprints, nor a significant interaction between these factors. For P_max_, there was a significant main effect of group (F (1, 117) = 4.65, *p* = 0.033, partial η^2^ = 0.03), with higher values observed in SPRs, while the main effect of number of sprints and the interaction between group and number of sprints were not significant.

Figure 2 shows the changes in the mechanical variables of the 1st sprint, 4th sprint, and 8th sprint of the two groups.

Multiple linear regression analysis was performed to investigate the relationship between slope decrement and the predictors vVO_2_max and ASR. The model accounted for a significant proportion of the variance in slope decrement (adjusted R^2^ = 0.35, F (2, 13) = 5.04, *p* = 0.024). vVO_2_max was a significant negative predictor of slope decrement (β = −4.26, t = −2.34, *p* = 0.036), whereas ASR did not reach significance. Specifically, higher values of vVO2max were associated with lower values of slope decrement. Formal diagnostics confirmed no significant violations of normality (Shapiro–Wilk W = 0.923, *p* = 0.189) or homoscedasticity (Breusch–Pagan *p* = 0.781, ncvTest *p* = 0.704), supporting the model’s validity. The residuals of the model were reasonably distributed, with no severe violations of normality or homoscedasticity, also supporting the model’s validity. Since ASR is defined as v_max_–vVO_2_max, we assessed multicollinearity and found VIFs of 7.79 for both predictors—below the conventional cutoff of 10—indicating that including both variables was acceptable despite their high correlation.

The repeated measures ANOVA for lactate showed that the main effect for group (F (1, 13) = 7.529, *p* = 0.0167, partial η^2^ = 0.37) and the repeated lactate measurements over time (F (4, 46) = 96.576, *p* < 0.001, partial η^2^ = 0.89) were statistically significant, as well as the interaction (F (3, 46) = 6.141, *p* = 0.0013, partial η^2^ = 0.29) (Figure 3). The post hoc analysis following the repeated measures ANOVA revealed significant differences between the MDR and SPR groups across multiple time points. For the baseline, there was no significant difference between the groups. However, significant differences emerged starting from the 4th sprint, where the MDRs showed lower lactate values compared to the SPRs (estimate = −6.80, t (13.8) = −4.156, *p* < 0.001). This trend continued and became more pronounced in subsequent sprints, with the largest differences observed at the 6th sprint (estimate = −7.06, t (13.8) = −4.319, *p* < 0.001) and 8th sprint (estimate = −7.26, t (13.8) = −3.626, *p* < 0.05). No comparison was possible for the 10th sprint due to non-estimable values.

## 4. Discussion

This study compared F–v parameters and aerobic capacity during repeated 10 × 60 m sprints for SPRs and MDRs, demonstrating that higher aerobic capacity in MDRs confers superior fatigue resistance without significant differences in v_max_. Key findings indicate that while MDRs maintained more stable sprint times, both groups exhibited comparable top-speed capabilities. The very large effect sizes for the aerobic variables between groups highlight potential differences, but these should be interpreted cautiously given the small sample. As we did not measure any other neuromuscular variables, we cannot speculate about additional mechanisms underlying this observation; however, SPRs, who rely more on anaerobic energy systems for short bursts of speed, experienced a sharp drop in sprint performance time and were unable to complete the total number of sprints required (i.e., five to eight instead of ten).

Although SPRs showed superior sprint performance in terms of F-v variables, there were no significant differences. Interestingly, there were no significant differences in v_0_ or v_max_ between the two groups, although the effect sizes were d = 0.76 and d = −0.96, respectively, suggesting practical differences. Such differences could maybe underpin sustained near-maximal velocity in SPRs for the 400 m and provide critical ASR for tactical surges and the end spurt in MDRs for the 800 m [24]. Additionally, earlier research highlights a large correlation between v_max_ and 800 m performance [25], as well as the importance of v_max_ for the 400 m, with vVO_2_max being a secondary predictor [26]. Nevertheless, these effects should be interpreted with caution given the sample size and variability. This finding does not align with earlier research showing that world-class 400 m athletes had a higher v_max_ compared with elite-level 800 m runners [25,27]. The moderate-to-large effect sizes for F_0_, P_max_, and RF_max_ suggest potential group differences in force production and power output, whereas the smaller and non-significant effects for DRF warrant cautious interpretation.

The MDRs’ higher VO_2_max reflects their superior aerobic conditioning, enabling better oxygen delivery and use during high-intensity sprints [28]. This finding fits with previous research, which suggests that aerobic capacity helps athletes recover more quickly between sprints, possibly because of faster PCr resynthesis [29]. Based on the multiple regression analysis, the observed relationship may reflect the influence of vVO_2_max on fatigue as significant predictors of slope decrement, with higher values of vVO_2_max associated with a lower decline in sprint performance time, which is supported by the faster PCr resynthesis of endurance-trained individuals [30]. This suggests that athletes with higher aerobic capacity experienced less fatigue, though this relationship is influenced by the inclusion of both groups in the analysis. Consequently, MDRs had more stable sprint times, showing their greater ability to resist fatigue [31].

The 10 × 60 m sprint test highlighted clear differences in the fatigue patterns of SPRs and MDRs. SPRs had a 16.6% fatigue slope decrement compared to MDRs’ 2.5%, indicating a faster decline in performance. SPRs were unable to complete the entire test, stopping after the 5th–8th sprint due to excessive fatigue, while MDRs completed all 10 sprints with stable times. SPRs, who started with faster times in the initial sprint, experienced a marked increase in sprint performance time with each successive sprint, indicating a decline in performance due to accumulated metabolic fatigue, as indicated by the high blood lactate values. By the 5th sprint, their sprint performance times approached those of the MDRs and afterwards, they could not complete the remaining sprints. Conversely, MDRs began with slower sprint times but maintained more consistent performance across repeated sprints, with less fluctuations in their times. This consistency suggests that they are better equipped to handle repeated high-intensity efforts, likely due to superior fatigue resistance and aerobic capacity.

Fatigue in SPRs during repeated-sprint efforts may be influenced by energy system reliance [32]. SPRs, whose training focuses on maximizing power and speed in short bursts, rely heavily on their anaerobic energy systems [14]. These systems provide immediate energy but lead to rapid accumulation of fatigue-inducing by-products, while the ability to resynthesize PCr may be lower, due to the lower aerobic capacity compared with the MDRs [25,29]. On the other hand, MDRs rely more on aerobic metabolism [1] and less on glycolysis, as indicated by the lower blood lactate concentration during the RS test. As a result, they exhibit lower performance decrements and faster recovery, as their aerobic system is more adept at replenishing ATP through oxidative pathways [14].

SPRs could not complete the test and managed to perform only five to eight sprints before volitional fatigue. Blood lactate concentration showed a steeper increase during the test, reaching values up to 20 mmol of lactate in the blood. MDRs, on the other hand, with their higher aerobic capacity, reached blood lactate values up to 15 mmol. This finding can be attributed to differences in anaerobic glycolysis and buffering capacity between groups, with SPRs demonstrating high rates of blood lactate accumulation, but an inability to tolerate it, and MDRs showing lower blood lactate, possibly due to both a lower glycolytic contribution and more effective clearance from the blood [33,34]. Furthermore, mechanical variables were different between groups. MDRs, while starting with a lower v_0_, showed a less pronounced decline, maintaining more consistent velocity over the course of the sprints. A similar study [18] focused on the effects of repeated sprints on elite male rugby players and their drop in v_0_ was coupled with a substantial decrease in P_max_, which fell by 20.1%. Similarly, the SPRs in the present study experienced notable fatigue during the 10 × 60 m sprint test, with sprint times increasing and force and P_max_ decreasing (20.3%), particularly after the 5th sprint. The sensitivity of P_max_ and v_0_ during RS has also been reported previously in another research [35], and it lends further support to the notion that RS efforts predominantly alter the mechanical variables of the horizontal F-v profile in speed-oriented athletes. In contrast, MDRs maintained their ability to apply force and velocity more consistently. Another key finding of the previous research [18] was that v_0_ was more affected by fatigue than F_0_ and P_max_. This aligns with the findings of this study in which SPRs’ v_0_ was greatly affected by fatigue during repeated sprints; the interaction of the group and number of sprints had a partial η^2^ = 0.24.

The multiple linear regression analysis highlighted the relationship between aerobic capacity and fatigue. vVO_2_max was a significant predictor of slope decrement, with higher values of vVO_2_max associated with lower fatigue-related declines. The stronger negative influence of vVO_2_max underscores the importance of aerobic capacity in mitigating fatigue during repeated sprints. Interestingly, athletes with a higher ASR exhibited greater performance drops. In this study, ASR was not significantly correlated with fatigue indices such as slope decrement, which is consistent with findings from other research showing no correlation with the fatigue index during repeated-sprint ability tests [36]. Even though ASR is used in high-intensity training, there is a gap in the literature about how it affects RS performance and sprint training, highlighting the need for further research to clarify its role and potential mechanisms in fatigue and performance adaptations. In contrast, MDRs, with superior aerobic capacity, demonstrated greater fatigue resistance, enabling them to sustain performance. While SPRs typically exhibit a higher ASR due to their lower vVO_2_max, a relatively larger ASR in MDRs could still provide benefits by allowing higher intensities above vVO_2_max, potentially influencing pacing strategies and tactical surges in middle-distance events [24]. Also, earlier research suggests that while ASR is often emphasized in high-intensity training, its interaction with aerobic capacity may critically influence fatigue resistance and RS performance, particularly highlighting differences between sprint- and endurance-oriented athletes [26]. These findings emphasize the critical role of aerobic capacity in recovery and in maintaining consistent performance during repeated high-intensity efforts. 

## 5. Conclusions

This study highlights the distinct differences in aerobic capacity and fatigue resistance between SPRs and MDRs. Middle-distance runners had higher vVO_2_max values and were able to maintain consistent performance during repeated-sprint tests, and unchanged F_0_, P_max_, v_0,_ and v_max_ during RS. In contrast, SPRs experienced greater fatigue during RS, despite showing better initial sprint performance.

The findings suggest that aerobic capacity, reflected in a higher vVO_2_max, may play an important role in mitigating the decline in sprint performance and mechanical variables changes during RS efforts and especially in v_0_. The large effect sizes despite the small sample size support the notion that athletes with higher aerobic capacity can better maintain repeated-sprint performance and v_0_ across repeated sprints compared with SPRs. The lack of significant changes in F_0_ and P_max_ suggest that these mechanical variables may not be as susceptible to fatigue during repeated-sprint efforts. Practically, coaches should integrate aerobic conditioning sessions for SPRs, as improved VO_2_max is linked to a smaller decrease in speed during the final 100 m, enhanced tolerance of metabolic acidosis, and better recovery [37,38], with interval training at 90–100% of VO_2_max being the most effective method [39,40]. MDRs may benefit from focused force and power development to optimize their sprint capabilities. The generalizability of these findings, however, is constrained by sample limitations (small size, male-only, and restricted to a single country). Future research should more precisely examine how aerobic capacity influences mechanical variables during RS across different populations by including female athletes, other sports, and adopting longitudinal designs to capture training-induced adaptations over time. 

## Figures and Tables

**Figure 1 jfmk-10-00342-f001:**
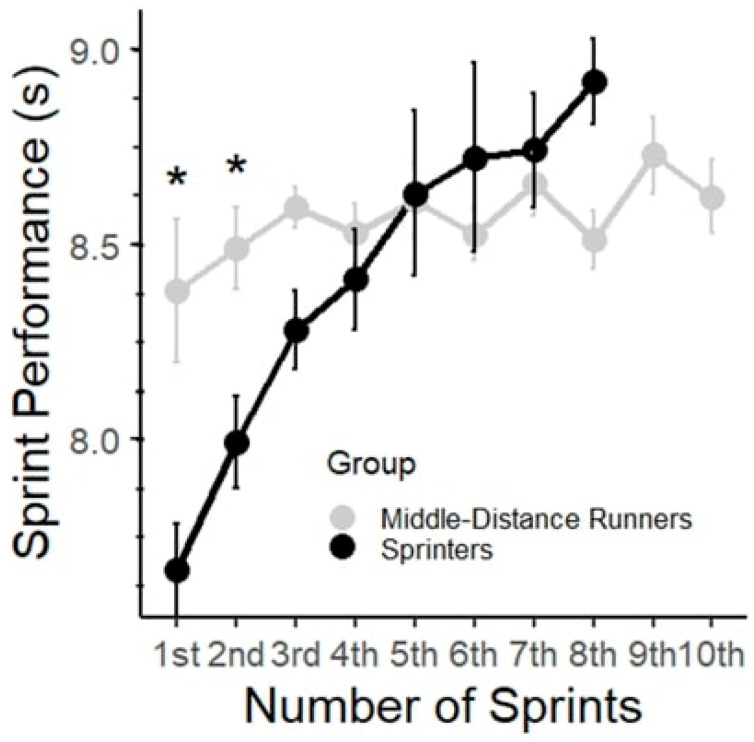
Performance time (s) for each 60 m sprint in the two groups. Data are presented as mean ± SD. * represents statistically significant differences between groups (*p* < 0.05).

**Figure 2 jfmk-10-00342-f002:**
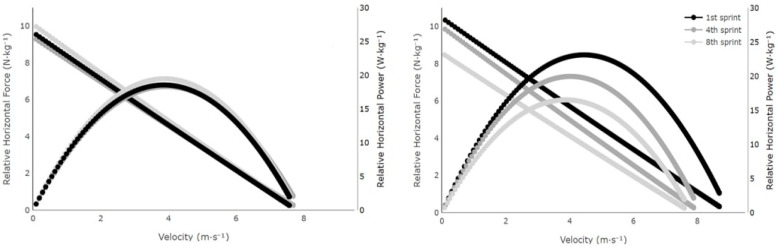
Changes in relative horizontal force, velocity, and relative horizontal power over the course of the sprints for the mean of middle-distance runners (**left** panel) and sprinters (**right** panel) compared to the 1st with the 4th and 8th.

**Figure 3 jfmk-10-00342-f003:**
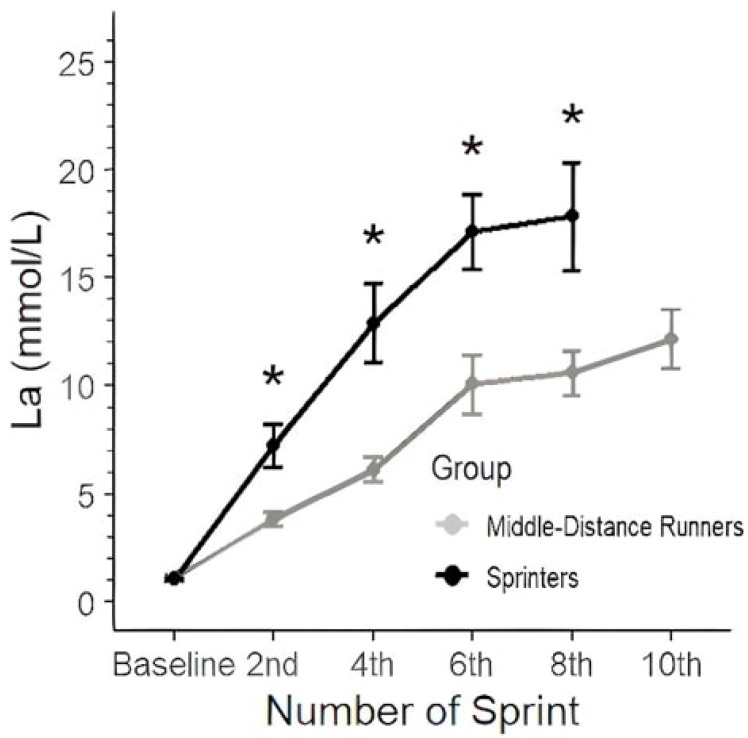
Blood lactate responses (mmol·L^−1^) during the 10 × 60 m sprint protocol for the two groups. Data are presented as mean ± SD. (*p* < 0.05) between groups at the corresponding sprint repetition. * represents statistically significant differences between groups.

**Table 1 jfmk-10-00342-t001:** Anthropometric characteristics and aerobic capacity characteristics. Values are presented as mean ± standard deviation.

Variable	Middle-Distance Runners (*n* = 8)	Sprinters (*n* = 8)	% Difference	*p* Value (Cohen’s d)
Height (cm)	179.0 ± 5.09	177.0 ± 4.3	1.1%	0.35 (0.48)
Weight (kg)	67.2 ± 5.01	74.0 ± 5.04	10.1%	0.02 (–1.33)
Body Fat (%)	8.07 ± 2.99	10.70 ± 1.8	32.4%	0.054 (–1.08)
V_max_ (m/s)	9.21 ± 0.412	9.72 ± 0.57	5.5%	0.06 (–1.03)
400 m (s)	52.2 ± 1.47	50.7 ± 1.00	2.9%	0.03 (1.21)
VO_2_max (mL/kg/min)	64.3 ± 3.28	55.4 ± 2.95	13.8%	<0.001 (2.86)
vVO_2_max (km/h)	20.0 ± 0.915	16.5 ± 0.78	17.5%	<0.001 (4.04)
VT (km/h)	15.7 ± 1.14	12.9 ± 0.55	17.8%	<0.001 (3.18)
ASR (km/h)	13.2 ± 1.37	18.5 ± 2.20	40.2%	<0.001 (–2.88)

V_max_ = maximum velocity; VO_2_max = maximal oxygen uptake; vVO_2_max = velocity at VO_2_max; VT = ventilatory threshold; ASR = anaerobic speed reserve.

**Table 2 jfmk-10-00342-t002:** Mechanical variables. Values are presented as mean ± standard deviation (SD).

Variable	Middle-Distance Runners (*n* = 8)	Sprinters (*n* = 8)	% Difference	*p* Value (Cohen’s d)
F_0_ (N/kg)	7.54 ± 2.27	8.31 ± 1.23	10.2%	0.41 (–0.42)
v_0_ (m/s)	9.09 ± 0.714	9.47 ± 0.67	4.2%	0.28 (–0.55)
P_max_ (W/kg)	16.9 ± 4.05	19.6 ± 2.45	16.0%	0.13 (–0.80)
RF_max_ (%)	43.45 ± 3.98	46.04 ± 2.97	6.0%	0.16 (–0.74)
DRF (%)	−7.90 ± 2.90	−7.99 ± 1.79	1.1%	0.94 (0.04)

F_0_ = maximal theoretical horizontal force; v_0_ = maximal theoretical horizontal velocity; P_max_= maximal theoretical horizontal power; RF_max_= maximal horizontal ratio of forces; DRF = decrease in ratio of force.

## Data Availability

The data are available upon request to the corresponding author.

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
