# Peer review of "Impact of Aerobic Capacity on Mechanical Variables in Track Sprinters and Middle-Distance Runners: A Comparative Study"

_jfmk, 2025, doi:10.3390/jfmk10030342_

Round 1

Reviewer 1 Report

Comments and Suggestions for Authors

Dear editor and authors, 

The manuscript presented here addresses the influence of aerobic capacity on the mechanical variables of sprints in middle-distance and sprint athletes. Although the manuscript presents a relevant topic, but the argumentative structure is confusing, as the logic behind investigating mechanical variables as proposed by the authors is unclear. Furthermore, the lack of information regarding neuromuscular responses during sprints weakens the presentation of the problem in the introduction and leaves the discussion lacking further clarification on how mechanical variables affect sprint performance. Attached is the PDF with my considerations to help improve the proposed work.

I remain available for further review if necessary.

Author Response

Line 42–43. Comment 1. Where are the neuromuscular factors? This only covers fiber type composition. I believe it's important to relate these responses to measures such as spinal reflexes and muscle activation, which also impact the neuromuscular system's responses.

Response 1: Thank you for your valuable comment. Accordingly, we have added a small reference about basic neuromuscular factors, as you have suggested in the manuscript. This change can be found on page [2], lines [48 - 51].

Line 44. Comment 2 .such as...?

Response 2: We agree with this observation. Accordingly, we have made the suggested revision by adding the two variables associated with sprint performance the most. This change can be found on page [2], lines [51]

Line 46–53. Comment 3.Here you need to explain more clearly which mechanical variables can interfere with anaerobic capacity performance, and how they affect sprint performance.

Response 3: Thank you for your valuable comment. We agree with this observation and combined it with the previous comment, and added a reference for this purpose This change can be found on page [2], lines [52 - 53]

Line 54–57. Comment 4.Similar to the previous comment. It needs to be more clear about the mechanical factors related to sprint performance that have been underexplored.

Response 4: Thank you for your valuable comment. Accordingly, we have added a small research gap fact, to our knowledge in the manuscript. This change can be found on page [2], lines [53 - 54]

Line 58–62. Comment 5. Again...mechanical factors. Explain with more details how to these factors affect the sprint performance.

Response 5: Thank you for your valuable comment. We agree with this observation and provided the more interesting variables. This change can be found on page [2], lines [71]

Line 63. Comment 6. Similar to mechanical factors. We need a more details about this.

Response 6: Thank you for your valuable comment. We have added the condition of the athletes, while fatigued to explain that possible influence. This change can be found on page [2], lines [73]

Line 82–86. Comment 7. Why ? What is your suggestion? Effects in mechanical factors ? Please explain better

Response 7: Thank you for your valuable comment. We agree with this observation. Accordingly, we have clarified this point in the manuscript. This change can be found on page [2], lines [77 - 79]

Line 144–151. Comment 8.Ok. But you measure lactate and dont talk aything ? You need to argue why you measure lactate and your importance to compheensive analysis to verify sustain sprint consistency.

Response 8: Thank you for your valuable comment. We agree with this observation. Accordingly, we added additional explanation in the introduction in the manuscript, explaining how it was used for as an indicator of metabolic intensity during the repeated sprints. This change can be found on page [2], lines [66 - 68]

Line 134–137. Comment 9. These variables need to explain in your introduction because you used this to construct your logical approach.

Response 9: Thank you for your valuable comment. We agree with this observation. Accordingly, we have added additional explanation in the manuscript. This change can be found on page [1], lines [39 - 41]

Line 182–200. Comment 10. Why you measure this variable ?You not argue to use lactate responses to explain mechanical factors or f-v parameters in sprint performance.

Response 10: Thank you for your valuable comment. We agree with this observation. Accordingly, we have made the suggested revision and clarified this in the manuscript. This change can be found on page [2], lines [39 - 43]

Line 154–166. Comment 11. I think so to bonferroni post hoc are more rigorous to your analysis. Please explain or reconsider your post hoc approach.

Response 11: Thank you for your valuable comment. We agree with this observation. Accordingly, we did the analysis using Bonferroni adjusted pairwise comparisons. Fortunately, the p-values of the already significant comparisons did not reach significance. This change can be found on page [4], lines [189 – 191]

Line 202–210. Comment 12. You cite tukey HSD for a post hoc test used. Please check in all statistical analysis and change if you need it.

Response 12: Thank you for your valuable comment. The previous Tukey HSD was used for the repeated ANOVAs, but this adjustment was for the MANOVA, which needs to be adjusted based on the number of variables it used. Of course, based on your review we changed to Bonferroni to have a more rigorous analysis.

Line 312–314. Comment 13. Please reconsider. Okay. I understood your argument. But, neuromuscular factors and f-v parameters were not addressed here. Are you sure there is no contribution from these variables?

Response 13: Thank you for your valuable comment. We agree with this observation, however we did not measure any neuromuscular factors, like the ones you suggested, we cannot discuss it. We added this comment in the text for the readers to speculated the absence of the underlying mechanisms. This change can be found on page [9], lines [326 - 330]

Line 321–326. Comment 14. You can talk about contact times, reactive strenght index and complementary viariabels associated with f-v parameters.

Response 14: Thank you for your valuable comment. Based on the feedback from another reviewer, we have chosen to keep the text concise and focused on the core topic of our study, limiting the discussion of additional variables that could influence the outcomes. This approach ensures that the manuscript remains coherent and centered on the research question we specifically investigated.

Line 331–333. Comment 15. Ok, but how these fact are associated with PCr resynthesis. Please you need more information to explain this.

Response 15: Thank you for your valuable comment. We agree with this observation. Accordingly, we have added an additional reference which strengthens the relations of aerobic variables and PCr resynthesis in the manuscript. This change can be found on page [9], lines [346 - 347]

Line 363–369. Comment 16. You can further explore the impacts on mechanical variables and F-V parameters. See these references for further information: 10.3390/ijerph20010704 ,10.1016/j.jbiomech.2011.07.020 AND 10.1007/s00421-020-04500-8.

Response 16: Thank you for your valuable comment and help of finding the right references. We used some of them to emphasize the sensitivity of two mechanical variables during repeated sprints in speed oriented athletes. This change can be found on page [10], lines [387 – 390]

Line 397–399. Comment 17. Again, this topic needs to be discussed and presented more clearly and in depth because the information is confusing due to the findings and comparisons that have been proposed.

Response 17: Thank you for your valuable comment. Based on our results, we concluded that, despite differences in aerobic capacity, the data do not indicate substantial susceptibility to fatigue during repeated sprints. We have clarified this in the revised manuscript and emphasized the importance of the findings, while noting that further research is needed to explore these relationships in more depth

Reviewer 2 Report

Comments and Suggestions for Authors

I hope this letter finds you well. I had the opportunity to review your article titled, “Impact of aerobic capacity on mechanical variables in track sprinters and middle-distance runners: A comparative study”, which was submitted the Journal of Functional Morphology and Kinesiology.

  1. Introduction

- They mentioned that sprinters (SPR) and middle-distance runners (MDR) differ in their energy system characteristics and suggested the rationale for comparing the two groups.

- The research gap was stated by stating that ‘the influence of ASR on RS mechanical response has not yet been sufficiently explored’.

- The purpose of the study was clearly stated at the end of the introduction: ‘To compare the influence of F–v variables and aerobic capacity on the performance of RS in sprinters and middle-distance runners.’

- Also, by directly presenting the hypothesis (“MDR will be less affected by RS performance than SPR due to its higher aerobic capacity”), the direction of the study is clear.

However, due to a lack of critical review of existing research, this study merely provides a simple literature list. → It is hoped that this study will reinforce the need for further research, as it presents conflicting results with recent research.

- The introduction is somewhat lengthy, reducing readability of the core topic (aerobic capacity vs. mechanical variables of RS performance).

  1. Method

- The study explicitly adopted a ‘comparative design’, dividing sprinters (SPR) and middle-distance runners (MDR) into two groups and analyzing RS performance and F–v profiles.

- It is appropriate to conduct three key tests, VO₂max, F–v profile, and RS test, 48 hours apart to minimize interference with post-exercise recovery.

- Some degree of external factor control was achieved by providing standardized warm-up and pre-conditioning (training, caffeine restriction, etc.).

- The fact that we conducted a ‘preliminary power analysis (G*Power)’ to calculate the sample size is an important factor that increases the validity of the study.

- The researcher attempted to confirm group homogeneity by presenting group-specific game records (400m, 800m) and age and physical conditions.

- VO₂max measurement uses the gold standard Douglas Bag method, ensuring high data accuracy.

- F–v profiles were obtained using a high-speed camera and validated methods to obtain reliable mechanical parameters.

- The RS test (10 × 60m, 30s recovery) simulates actual sports situations and has the advantage of integrating analysis of metabolic indicators, including blood lactate measurement.

- Appropriate analysis procedures such as MANOVA, repeated measures ANOVA, and regression analysis were used for group comparison.

- Report effect sizes (Cohen’s d, partial η²) to provide more information than just a simple significance test.

However, in the RS test, if some sprinters failed to complete 10 sprints (stopped 5-8 times), the missing value handling method described only as "missing value removal (omit)" poses a risk of data bias. Conservative approaches (e.g., mixed-effects models, multiple imputation) were not considered.

- There is no discussion of the possibility that a ‘learning effect’ or cumulative fatigue may have an effect in performing all of the VO₂max, F–v, and RS tests.

Despite the small sample size, applying "multivariate analysis (MANOVA, regression)" poses a risk of overfitting. In particular, predictor multicollinearity (VIF=7.79) was identified in the regression analysis, but simply stating that this was "within the acceptable range" is unconvincing. Further explanation is needed.

  1. Results

- Key characteristics (physical condition, VO₂max, vVO₂max, ASR, etc.) by group (sprinters vs. middle distance runners) are clearly presented in the table (Table 1).

- The results range is wide as it is analyzed step by step including F–v profile and RS test results and blood lactate concentration.

- High transparency by intuitively presenting results through statistical test values (F value, p value, effect size) and graphs (Figure 1–3).

- It was clearly demonstrated that MDR was significantly higher than SPR in VO₂max, vVO₂max, and VT, and also had higher fatigue resistance in the RS test.

- The phenomenon of SPR voluntary withdrawal during RS is an experimentally interesting finding and also has practical implications.

- It is judged that the academic contribution is high in that vVO₂max was derived as a major predictor of fatigue resistance through multiple regression analysis.

- However, the text is overflowing with detailed figures and p-values, obscuring the core message. International academic journals generally recommend presenting "representative figures + direction of effect," with detailed figures placed in a table (appendix).

- Some interpretations overemphasize statistically insignificant results. For example, in v₀, F₀, Pₘₐₓ, etc., the p-value is insignificant, but claiming a "potentially substantial difference" based on the effect size is scientifically unconvincing in situations with extremely small sample sizes.

  1. Discussion

- The discussion well interprets the key finding that middle-distance runners (MDR) showed higher fatigue resistance in repeated sprints (RS) compared to sprinters (SPR).

- Emphasizes that vVO₂max is a key predictor of RS performance decline, directly addressing the research question.

- The interpretation of the results is theoretically based on the differences between the two groups, which are explained based on physiological mechanisms such as differences in energy systems (anaerobic vs. aerobic), lactate accumulation, and PCr resynthesis.

- The paper's strength lies in its practical implications: "SPRs also need to strengthen their aerobic training" and "MDRs need strength and power training." These implications provide a message directly applicable to coaching and training settings.

- It has a convergent contribution in that existing RS studies rarely address mechanical variables (F–v profile) and metabolic indices (lactate, vVO₂max) simultaneously.

- However, the practical implications have been described somewhat superficially. Providing guidance on specific approaches needed in training program design would enhance practicality.

- While emphasizing the research contribution, it does not sufficiently acknowledge the difficulties in generalizing

due to ‘sample limitations (small size, male only, single country)’.

- Future research suggestions also remain at the level of simply ‘additional research needed’ and lack specific suggestions (e.g., inclusion of female athletes, application to various sports, application of longitudinal design, etc.).

Author Response

Comment 1. They mentioned that sprinters (SPR) and middle-distance runners (MDR) differ in their energy system characteristics and suggested the rationale for comparing the two groups.

Response 1: Thank you for your valuable comment. Your positive comments are encouraging.

Comment 2. The research gap was stated by stating that ‘the influence of ASR on RS mechanical response has not yet been sufficiently explored’.

Response 2: Thank you for your valuable comment. Your positive comments are encouraging.

Comment 3. The purpose of the study was clearly stated at the end of the introduction: ‘To compare the influence of F–v variables and aerobic capacity on the performance of RS in sprinters and middle-distance runners.’

Response 3: Thank you for your valuable comment. Your positive comments are encouraging.

Comment 4. Also, by directly presenting the hypothesis (“MDR will be less affected by RS performance than SPR due to its higher aerobic capacity”), the direction of the study is clear.

Response 4: Thank you for your valuable comment. Your positive comments are encouraging.

Comment 5. However, due to a lack of critical review of existing research, this study merely provides a simple literature list. → It is hoped that this study will reinforce the need for further research, as it presents conflicting results with recent research.

Response 5: Thank you for your valuable comment. We are trying to create more variables to control training regimens for runners

Comment 6. The introduction is somewhat lengthy, reducing readability of the core topic (aerobic capacity vs. mechanical variables of RS performance).

Response 6: Thank you for your valuable comment. We agree with this observation. Based on other reviewers comments, we added and subtracted information to make it more concise.

Comment 7. The study explicitly adopted a ‘comparative design’, dividing sprinters (SPR) and middle-distance runners (MDR) into two groups and analyzing RS performance and F–v profiles.

Response 7: Thank you for your valuable comment.

Comment 8. It is appropriate to conduct three key tests, VO₂max, F–v profile, and RS test, 48 hours apart to minimize interference with post-exercise recovery.

Response 8: Thank you for your valuable comment. Your positive comments are encouraging.

Comment 9. Some degree of external factor control was achieved by providing standardized warm-up and pre-conditioning (training, caffeine restriction, etc.).

Response 9: Thank you for your valuable comment.

Comment 10. The fact that we conducted a ‘preliminary power analysis (G*Power)’ to calculate the sample size is an important factor that increases the validity of the study.

Response 10: Thank you for your valuable comment. Your positive comments are encouraging.

Comment 11. The researcher attempted to confirm group homogeneity by presenting group-specific game records (400m, 800m) and age and physical conditions.

Response 11: Thank you for your valuable comment. Your positive comments are encouraging.

Comment 12. VO₂max measurement uses the gold standard Douglas Bag method, ensuring high data accuracy.

Response 12: Thank you for your valuable comment. Your positive comments are encouraging.

Comment 13. F–v profiles were obtained using a high-speed camera and validated methods to obtain reliable mechanical parameters.

Response 13: Thank you for your valuable comment. Your positive comments are encouraging.

Comment 14. The RS test (10 × 60m, 30s recovery) simulates actual sports situations and has the advantage of integrating analysis of metabolic indicators, including blood lactate measurement.

Response 14: Thank you for your valuable comment.

Comment 15. Appropriate analysis procedures such as MANOVA, repeated measures ANOVA, and regression analysis were used for group comparison.

Response 15: Thank you for your valuable comment. Your positive comments are encouraging.

Comment 16. Report effect sizes (Cohen’s d, partial η²) to provide more information than just a simple significance test.

Response 16: Thank you for your valuable comment.

Comment 17. However, in the RS test, if some sprinters failed to complete 10 sprints (stopped 5-8 times), the missing value handling method described only as "missing value removal (omit)" poses a risk of data bias. Conservative approaches (e.g., mixed-effects models, multiple imputation) were not considered.

Response 17: Thank you for your valuable comment. In this study, multiple imputation methods were not applied because they would not provide accurate estimations for sprint performance data. The repeated sprint test required split times at several distances (5–60 m), making it inappropriate to artificially generate values for missing performances. Using imputation in this context could lead to unrealistic outcomes, such as predicting excessively long times (e.g., 15 seconds over 60 m), which would not represent actual sprint ability and would introduce further bias into the analysis. Similarly, mixed-effects models were not employed, as the primary aim was to use a straightforward analytical approach that would allow for clear interpretation of group and sprint comparisons. While mixed-effects models are valuable for handling complex datasets with missing values, their additional complexity was not aligned with the focus of this analysis, which prioritized transparency and simplicity in reporting performance outcomes.

Comment 18. There is no discussion of the possibility that a ‘learning effect’ or cumulative fatigue may have an effect in performing all of the VO₂max, F–v, and RS tests.

Response 18: Thank you for your valuable comment. They were all trained runners who regularly performed all the tests used in this study as part of their annual training routine Accordingly, we clarified this point in the manuscript. This change can be found on page [2-3], lines [90 - 92]

Comment 19. Despite the small sample size, applying "multivariate analysis (MANOVA, regression)" poses a risk of overfitting. In particular, predictor multicollinearity (VIF=7.79) was identified in the regression analysis, but simply stating that this was "within the acceptable range" is unconvincing. Further explanation is needed.

Response 19: Thank you for your valuable comment. We agree with this observation. Based on another reviewers comment we enforced they analysis by providing Shapiro-Wilks and Breusch-Pagan and ncvTest values supporting the models validity. This change can be found on page [8], lines [294 - 297]

Comment 20. Key characteristics (physical condition, VO₂max, vVO₂max, ASR, etc.) by group (sprinters vs. middle distance runners) are clearly presented in the table (Table 1).

Response 20: Thank you for your valuable comment.

Comment 21. The results range is wide as it is analyzed step by step including F–v profile and RS test results and blood lactate concentration.

Response 21: Thank you for your valuable comment. We agree with this observation. They are wide and that is why we tried to concentrate our data to only the significant, as you suggest in comment 26.

Comment 22. High transparency by intuitively presenting results through statistical test values (F value, p value, effect size) and graphs (Figure 1–3).

Response 22: Response 1: Thank you for your valuable comment. Your positive comments are encouraging.

Comment 23. It was clearly demonstrated that MDR was significantly higher than SPR in VO₂max, vVO₂max, and VT, and also had higher fatigue resistance in the RS test.

Response 23: Thank you for your valuable comment. We agree with this observation.

Comment 24. The phenomenon of SPR voluntary withdrawal during RS is an experimentally interesting finding and also has practical implications.

Response 24: Thank you for your valuable comment. We agree with this observation.

Comment 25. It is judged that the academic contribution is high in that vVO₂max was derived as a major predictor of fatigue resistance through multiple regression analysis.

Response 25: Thank you for your valuable comment. Your positive comments are encouraging.

Comment 26. However, the text is overflowing with detailed figures and p-values, obscuring the core message. International academic journals generally recommend presenting "representative figures + direction of effect," with detailed figures placed in a table (appendix).

Response 26: Thank you for your valuable comment. We agree with this observation. Accordingly, we have removed the insignificant results from the Results chapter to make it easier for the reader. Also, we concise sentences and paragraphs to clarify the main message in the manuscript. This change can be found on page [6-7], lines [246 – 258, 266 -282]

Comment 27. Some interpretations overemphasize statistically insignificant results. For example, in v₀, F₀, Pₘₐₓ, etc., the p-value is insignificant, but claiming a "potentially substantial difference" based on the effect size is scientifically unconvincing in situations with extremely small sample sizes.

Response 27: Thank you for your valuable comment. We agree with this observation. Accordingly, we have made the suggested revision in the manuscript, to not overemphasize this insignificant statistically results. This change can be found on page [7], lines [266 -282]]

Comment 28. The discussion well interprets the key finding that middle-distance runners (MDR) showed higher fatigue resistance in repeated sprints (RS) compared to sprinters (SPR).

Response 28: Thank you for your valuable comment.

Comment 29. Emphasizes that vVO₂max is a key predictor of RS performance decline, directly addressing the research question.

Response 29: Thank you for your valuable comment. Your positive comments are encouraging.

Comment 30. The interpretation of the results is theoretically based on the differences between the two groups, which are explained based on physiological mechanisms such as differences in energy systems (anaerobic vs. aerobic), lactate accumulation, and PCr resynthesis.

Response 30: Thank you for your valuable comment. Your positive comments are encouraging.

Comment 31. The paper's strength lies in its practical implications: "SPRs also need to strengthen their aerobic training" and "MDRs need strength and power training." These implications provide a message directly applicable to coaching and training settings.

Response 31: Thank you for your valuable comment. Your positive comments are encouraging.

Comment 32. It has a convergent contribution in that existing RS studies rarely address mechanical variables (F–v profile) and metabolic indices (lactate, vVO₂max) simultaneously.

Response 32: Thank you for your valuable comment.

Comment 33. However, the practical implications have been described somewhat superficially. Providing guidance on specific approaches needed in training program design would enhance practicality.

Response 33: Thank you for your valuable comment. We agree with this observation. Based on another reviewers comment, as well as yours, we encourage coaches to integrate aerobic conditioning for sprinters. This change can be found on page [11], lines [427 - 430]

Comment 34. While emphasizing the research contribution, it does not sufficiently acknowledge the difficulties in generalizing due to ‘sample limitations (small size, male only, single country)’.

Response 34: Thank you for your valuable comment. We agree with this observation. Accordingly, we have made the suggested revision in the manuscript for the reader to understand its limitations. This change can be found on page [11], lines [431 – 433]

Comment 35. Future research suggestions also remain at the level of simply ‘additional research needed’ and lack specific suggestions (e.g., inclusion of female athletes, application to various sports, application of longitudinal design, etc.).

Response 35: Thank you for your valuable comment. We agree with this observation. Accordingly, we have made the suggested revision in the manuscript. This change can be found on page [11], lines [433 - 436]

Reviewer 3 Report

Comments and Suggestions for Authors

Article

Impact of Aerobic Capacity on Mechanical Variables in Track Sprinters and Middle-Distance Runners: A Comparative Study

General Comment to the Authors:

Thank you for submitting this manuscript. The study addresses a relevant topic with a solid methodology and contributes to our understanding of fatigue resistance and sprint mechanics. To improve clarity and scientific rigor, I’ve provided focused comments on statistical interpretation, physiological reasoning, and consistency across sections. I encourage you to revise key areas, especially regarding non-significant findings with large effect sizes, the role of ASR, and practical training implications.

Abstract (Lines 10–30)
Line 10. Comment 1: Correct the spelling of “Backround” to “Background”.

Line 15. Comment 2: Clarify whether “sprinting 2 × 60 m” refers to two maximal sprints or a specific protocol for mechanical F-v profiling.

Line 29. Comment 3: Clarify the statement “but not v₀ and Vmax,” as it appears inconsistent with earlier results indicating a significant decrease in v₀. Ensure alignment with the Results section.

Introduction (Lines 33–63)
Lines 33–41. Comment 4: Consider condensing the first paragraph, as several ideas are repeated or could be expressed more concisely. The concepts of RS ability, mechanical variables, and fatigue resistance are introduced in overlapping ways. A more streamlined presentation would improve readability and focus.

Lines 42–48. Comment 5: Clarify the distinction between VO₂max and vVO₂max earlier in the paragraph. While both are mentioned, the transition from VO₂max to vVO₂max could be more explicit, especially since the latter is emphasized later as a stronger predictor.

Lines 50–52. Comment 6: Elaborate briefly on why ASR remains underexplored in RS mechanics. A sentence indicating the lack of prior empirical studies or methodological challenges would strengthen the rationale.

Lines 54–56. Comment 7: Rephrase the sentence “SPR prioritizing anaerobic power, MDR emphasizing aerobic endurance” to improve grammatical flow. Suggested: “SPR typically prioritize anaerobic power, whereas MDR emphasize aerobic endurance.”

Materials and Methods (Lines 64–179)
Lines 65–69. Comment 8: Clarify whether the assumed effect size (f = 0.3) and correlation (r = 0.5) used in the power analysis were based on previous literature or pilot data. This would strengthen the justification of the sample size.

Lines 70–71. Comment 9: Clarify the rationale for including 400 m training time for MDR. Was this used to ensure comparable sprinting experience with SPR? A brief explanation would help contextualize the selection criteria.

Lines 88–114. Comment 10: Clarify whether the Douglas Bag method was used continuously or only during the final 30 seconds of each stage. Also, indicate whether the same technician conducted all VO₂max tests to minimize inter-rater variability.

Lines 117–130. Comment 11: Specify whether both 60 m sprints were averaged or analyzed separately when deriving F₀, v₀, and Pₘₐₓ. This affects the reliability of the mechanical profiling.

Lines 153–179. Comment 12: Clarify whether assumptions for repeated measures ANOVA (e.g., sphericity) were tested and, if violated, whether corrections (e.g., Greenhouse-Geisser) were applied. This would enhance the transparency of the statistical analysis.

Results (Lines 180–304)
Lines 184–185. Comment 13: Clarify the interpretation of the non-significant difference in Vmax (p = 0.06) given the large effect size (d = -1.03). This discrepancy may confuse readers and warrants a brief explanation regarding practical vs. statistical significance.

Lines 219–221. Comment 14: Clarify whether the broader range of fluctuations in v₀ for SPR reflects individual variability or a consistent fatigue pattern across participants. This would strengthen the interpretation of the interaction effect.

Lines 243–245. Comment 15: Clarify whether the fatigue slope decrement was calculated using all available sprints or only completed sprints per participant. This affects the comparability between SPR and MDR.

Lines 256–259. Comment 16: Specify whether the regression model was tested for residual normality and homoscedasticity using formal diagnostics (e.g., plots or tests), rather than stating “reasonably distributed.” This would enhance methodological transparency.

Discussion (Lines 305–383)
Lines 308–310. Comment 17: Clarify that the non-significant difference in Vmax does not imply equivalence, especially given the large effect size. This strengthens the interpretation of “comparable top-speed capabilities” between groups.

Lines 346–352. Comment 18: Deepen the discussion of lactate accumulation and fatigue by referencing physiological mechanisms such as buffering capacity and oxidative clearance. This would enhance the interpretation of group differences.

Lines 373–379. Comment 19: Briefly discuss why ASR, despite its theoretical relevance, did not emerge as a significant predictor in the regression analysis. This adds nuance to the interpretation of the findings.

Conclusions (Lines 385–403)
Lines 385–387. Comment 20: Soften the phrase “demonstrating superior fatigue resistance compared to SPR” to “suggesting superior fatigue resistance,” considering the small sample size and variability.

Lines 392–394. Comment 21: Revise the statement that “variables such as Vmax and v₀ were affected by aerobic capacity,” since Vmax did not show significant changes. Clarify the distinction between statistical and practical effects.

Lines 398–402. Comment 22: Specify what types of aerobic-conditioning sessions (e.g., interval-based, tempo runs) may be most beneficial for SPR. This would enhance the practical value of the recommendation for coaches.

Author Response

Comments Abstract (Lines 10–30)

Line 10. Comment 1: Correct the spelling of “Backround” to “Background”.

Response 1: Thank you for your valuable comment. We corrected our orthographical mistake. This change can be found on page [1], lines [10]

Line 15. Comment 2: Clarify whether “sprinting 2 × 60 m” refers to two maximal sprints or a specific protocol for mechanical F-v profiling.

Response 2: Thank you for your valuable comment. We agree with this observation. Accordingly, we clarified this point by adding the recovery which was full in the manuscript. This change can be found on page [1], lines [15]

Line 29. Comment 3: Clarify the statement “but not v₀ and Vmax,” as it appears inconsistent with earlier results indicating a significant decrease in v₀. Ensure alignment with the Results section.

Response 3: Thank you for your valuable comment. We agree with this observation. Accordingly, we have made the suggested revision of my mistake in the manuscript. This change can be found on page [1], lines [29 - 30]

Comments Introduction (Lines 33–63)

Lines 33–41. Comment 4: Consider condensing the first paragraph, as several ideas are repeated or could be expressed more concisely. The concepts of RS ability, mechanical variables, and fatigue resistance are introduced in overlapping ways. A more streamlined presentation would improve readability and focus.

Response 4: Thank you for your valuable comment. We agree with this observation. Accordingly, we have concised some ideas in the manuscript. Based on other reviewers comments we have tried to add and subtract and believe that there are not overlapping anymore This change can be found on page [2], lines [32,36, 38-43]

Lines 42–48. Comment 5: Clarify the distinction between VO₂max and vVO₂max earlier in the paragraph. While both are mentioned, the transition from VO₂max to vVO₂max could be more explicit, especially since the latter is emphasized later as a stronger predictor.

Response 5: Thank you for your valuable comment. We agree with this observation. Accordingly, we have clarified this point by adding additional explanation in the manuscript. This change can be found on page [2], lines [58- 61]

Lines 50–52. Comment 6: Elaborate briefly on why ASR remains underexplored in RS mechanics. A sentence indicating the lack of prior empirical studies or methodological challenges would strengthen the rationale.

Response 6: Thank you for your valuable comment. We agree with this observation. Accordingly, we have made the suggested revision by indicating the lack of prior studies to our knowledge in the manuscript. This change can be found on page [2], lines [62 - 65]

Lines 54–56. Comment 7: Rephrase the sentence “SPR prioritizing anaerobic power, MDR emphasizing aerobic endurance” to improve grammatical flow. Suggested: “SPR typically prioritize anaerobic power, whereas MDR emphasize aerobic endurance.”

Response 7: Thank you for your valuable comment. We agree with this observation. Accordingly, we have made the suggested revision. This change can be found on page [2], lines [70]

Comments Materials and Methods (Lines 64–179)

Lines 65–69. Comment 8: Clarify whether the assumed effect size (f = 0.3) and correlation (r = 0.5) used in the power analysis were based on previous literature or pilot data. This would strengthen the justification of the sample size.

Response 8: Thank you for your valuable comment. The assumed effect size (f = 0.3) and correlation (r = 0.5) used in our power analysis were based on commonly used conventions for repeated-measures designs

Lines 70–71. Comment 9: Clarify the rationale for including 400 m training time for MDR. Was this used to ensure comparable sprinting experience with SPR? A brief explanation would help contextualize the selection criteria.

Response 9: Thank you for your valuable comment. We agree with this observation. Accordingly, we have added additional explanation “The inclusion of 400 m training times for MDR was used to ensure comparable sprinting experience between groups, providing a relevant benchmark for performance comparisons” in the manuscript. This change can be found on page [5], lines [211 - 215]

Lines 88–114. Comment 10: Clarify whether the Douglas Bag method was used continuously or only during the final 30 seconds of each stage. Also, indicate whether the same technician conducted all VO₂max tests to minimize inter-rater variability.

Response 10: Thank you for your valuable comment. We agree with this observation. Accordingly, we have clarified this point in the manuscript. This change can be found on page [3], lines [131 - 134]

Lines 117–130. Comment 11: Specify whether both 60 m sprints were averaged or analyzed separately when deriving F₀, v₀, and Pₘₐₓ. This affects the reliability of the mechanical profiling.

Response 11: Thank you for your valuable comment. We agree with this observation. Accordingly, we have clarified this point in the manuscript. This change can be found on page [4], lines [150 - 151]

Lines 153–179. Comment 12: Clarify whether assumptions for repeated measures ANOVA (e.g., sphericity) were tested and, if violated, whether corrections (e.g., Greenhouse-Geisser) were applied. This would enhance the transparency of the statistical analysis.

Response 12: Thank you for your valuable comment. We agree with this observation. Accordingly, we have made the suggested revision in the manuscript. This change can be found on page [4], lines [181 - 183]

Comments Results (Lines 180–304)

Lines 184–185. Comment 13: Clarify the interpretation of the non-significant difference in Vmax (p = 0.06) given the large effect size (d = -1.03). This discrepancy may confuse readers and warrants a brief explanation regarding practical vs. statistical significance.

Response 13: Thank you for your valuable comment. We agree with this observation. Accordingly, we have added additional explanation, by emphasizing the meaningful practical difference, despite the statistical insignificance in the manuscript. This change can be found on page [5], lines [209 - 211]

Lines 219–221. Comment 14: Clarify whether the broader range of fluctuations in v₀ for SPR reflects individual variability or a consistent fatigue pattern across participants. This would strengthen the interpretation of the interaction effect.

Response 14: Thank you for your valuable comment. We agree with this observation. Accordingly, we have calculated the within participant CV and provided more information in our manuscript. This change can be found on page [7], lines [274 - 277]

Lines 243–245. Comment 15: Clarify whether the fatigue slope decrement was calculated using all available sprints or only completed sprints per participant. This affects the comparability between SPR and MDR.

Response 15: Thank you for your valuable comment. We agree with this observation. Accordingly, we have clarified this point in the manuscript, by emphasizing the completed sprints. Based on another reviewers comments this paragraph has changed, because it got concised with another. This change can be found on page [6], lines [256]

Lines 256–259. Comment 16: Specify whether the regression model was tested for residual normality and homoscedasticity using formal diagnostics (e.g., plots or tests), rather than stating “reasonably distributed.” This would enhance methodological transparency.

Response 16: Thank you for your valuable comment. We agree with this observation. Accordingly, we have added a Shapiro Wilk, a Breusch Pagan test to support its validity in the manuscript. This change can be found on page [8], lines [294 - 297]

Comments Discussion (Lines 305–383)

Lines 308–310. Comment 17: Clarify that the non-significant difference in Vmax does not imply equivalence, especially given the large effect size. This strengthens the interpretation of “comparable top-speed capabilities” between groups.

Response 17: Thank you for your valuable comment. We made it more consisted and clear of this interpretaion. This change can be found on page [9], lines [333 - 335]

Lines 346–352. Comment 18: Deepen the discussion of lactate accumulation and fatigue by referencing physiological mechanisms such as buffering capacity and oxidative clearance. This would enhance the interpretation of group differences.

Response 18: Thank you for your valuable comment. We agree with this observation. Accordingly, we have added additional explanation and references to add this abilities in the manuscript. This change can be found on page [10], lines [376 - 381]

Lines 373–379. Comment 19: Briefly discuss why ASR, despite its theoretical relevance, did not emerge as a significant predictor in the regression analysis. This adds nuance to the interpretation of the findings.

Response 19: Thank you for your valuable comment. We agree with this observation. Accordingly, we have clarified this point by adding reference in the manuscript. This change can be found on page [11], lines [400 - 402]

Comments Conclusions (Lines 385–403)

Lines 385–387. Comment 20: Soften the phrase “demonstrating superior fatigue resistance compared to SPR” to “suggesting superior fatigue resistance,” considering the small sample size and variability.

Response 20: Thank you for your valuable comment. We agree with this observation. Accordingly, we have made the suggested revision. This change can be found on page [11], lines [415]

Lines 392–394. Comment 21: Revise the statement that “variables such as Vmax and v₀ were affected by aerobic capacity,” since Vmax did not show significant changes. Clarify the distinction between statistical and practical effects.

Response 21: Thank you for your valuable comment. We agree with this observation. Accordingly, we have clarified this point like in the other parts of the manuscript, by emphasizing the difference of practical and statistical results. This change can be found on page [11], lines [420]

Lines 398–402. Comment 22: Specify what types of aerobic-conditioning sessions (e.g., interval-based, tempo runs) may be most beneficial for SPR. This would enhance the practical value of the recommendation for coaches.

Response 22: Thank you for your valuable comment. We agree with this observation. Accordingly, we have provided with the literature suggested training for SPR who train 400 m in the manuscript. This change can be found on page [11], lines [427 - 430]

Round 2

Reviewer 1 Report

Comments and Suggestions for Authors

Dear editor, 

Thank you again for the opportunity to review this paper. 

I am satisfied with the corrections made by the authors, and I recommend accepting the work for publication.

Best regards

Author Response

Thank you for your acceptance!

Reviewer 3 Report

Comments and Suggestions for Authors

Article

Impact of Aerobic Capacity on Mechanical Variables in Track Sprinters and Middle-Distance Runners: A Comparative Study

My Final Revision

Dear Authors,

Thank you for your thoughtful and constructive revisions to the manuscript. The updated version demonstrates significant improvement in clarity, methodological transparency, and physiological interpretation. Your efforts to address my previous comments are appreciated. Below are a few final minor suggestions to further refine the manuscript before publication:

Comment 1. Physiological Relevance of Vo and Vmax
See at discussion, second paragraph
The discussion of Vo and Vmax remains limited. I suggest elaborating on how these variables reflect aerobic power and how they may relate to performance demands in sprint versus middle-distance running. This would enhance the physiological depth of the manuscript.

Comment 2. Interpretation of Anaerobic Speed Reserve (ASR)
See at discussion, third paragraph
While the ASR is now better contextualized, I recommend expanding its interpretation by briefly explaining its relevance to the balance between aerobic and anaerobic energy systems. Consider discussing how ASR may influence pacing strategies or tactical decisions in middle-distance events.

Comment 3. Limitations of the Study
See at final paragraph of Discussion
The limitations section could be strengthened by including comments on sample size, gender distribution, and the absence of longitudinal or intervention data. This would provide a more balanced perspective on the generalizability of the findings.

Comment 4. Units of Measurement in Tables
See: Table 1 and Table 2
Please ensure that all units of measurement (e.g., m/s, ml/kg/min) are clearly indicated within the table columns. This will improve clarity and reduce ambiguity for readers.

Comment 5. Practical Applications of Findings
See at discussion, final third
While the manuscript includes useful practical suggestions, I recommend making the connection between specific findings (e.g., soleus stiffness, ASR) and training interventions more explicit. For example, which types of training sessions or modalities might target these adaptations?

Comment 6. Language Consistency and Terminology
Throughout the manuscript (if you willing to do it)
The language is generally clear, but I suggest a final proofreading to ensure consistency in terminology (e.g., “mechanical variables” vs. “performance metrics”) and to correct minor grammatical inconsistencies.

It is noteworthy to acknowledge the caliber of your work and your responsiveness throughout the review process.

Author Response

Comment 1. Physiological Relevance of Vo and Vmax
See at discussion, second paragraph
The discussion of Vo and Vmax remains limited. I suggest elaborating on how these variables reflect aerobic power and how they may relate to performance demands in sprint versus middle-distance running. This would enhance the physiological depth of the manuscript.

Response 1: Thank you for your valuable comment. Accordingly, we have added a text regarding the effectiveness of vmax for the 400 m and 800 m. This change can be found on page [9], lines [334 - 338].

Comment 2. Interpretation of Anaerobic Speed Reserve (ASR)
See at discussion, third paragraph
While the ASR is now better contextualized, I recommend expanding its interpretation by briefly explaining its relevance to the balance between aerobic and anaerobic energy systems. Consider discussing how ASR may influence pacing strategies or tactical decisions in middle-distance events.

Response 2: Thank you for your valuable comment. Accordingly, we have interpreted with a more practical point of view. This change can be found on page [11], lines [401 - 404].

Comment 3. Limitations of the Study
See at final paragraph of Discussion
The limitations section could be strengthened by including comments on sample size, gender distribution, and the absence of longitudinal or intervention data. This would provide a more balanced perspective on the generalizability of the findings.

Response 3: Thank you for your valuable comment. Based on the comments of another reviewer, we have added the limitations of the study as well as our indications regarding future research. This can be found on page [11], lines [412 - 418].

Comment 4. Units of Measurement in Tables
See: Table 1 and Table 2
Please ensure that all units of measurement (e.g., m/s, ml/kg/min) are clearly indicated within the table columns. This will improve clarity and reduce ambiguity for readers.

Response 4: Thank you for your valuable comment. Accordingly, we have made the suggested revision. This change can be found on page [6], Table 2.

Comment 5. Practical Applications of Findings
See at discussion, final third
While the manuscript includes useful practical suggestions, I recommend making the connection between specific findings (e.g., soleus stiffness, ASR) and training interventions more explicit. For example, which types of training sessions or modalities might target these adaptations?

Response 5: Thank you for your valuable comment. Based on previous comments we have added the practical applications about performance (Lines 434 – 439). ASR being the difference between vmax and vVO2max, training the can influence both is a way to influence ASR.

Comment 6. Language Consistency and Terminology
Throughout the manuscript (if you willing to do it)
The language is generally clear, but I suggest a final proofreading to ensure consistency in terminology (e.g., “mechanical variables” vs. “performance metrics”) and to correct minor grammatical inconsistencies.

Response 6: Thank you for your valuable comment. Accordingly, we made the ssuggested evision in different parts of manuscripts, correcting all mechanical outputs, parameters and others to mechanical variables.